# Genome-Scale Modeling Specifies the Metabolic Capabilities of *Rhizophagus irregularis*

Philipp Wendering,[a] Zoran Nikoloski[a,b]

[a]Bioinformatics, Institute of Biochemistry and Biology, University of Potsdam, Potsdam, Germany
[b]Systems Biology and Mathematical Modeling, Max Planck Institute of Molecular Plant Physiology, Potsdam, Germany

**ABSTRACT** *Rhizophagus irregularis* is one of the most extensively studied arbuscular mycorrhizal fungi (AMF) that forms symbioses with and improves the performance of many crops. Lack of transformation protocol for *R. irregularis* renders it challenging to investigate molecular mechanisms that shape the physiology and interactions of this AMF with plants. Here, we used all published genomics, transcriptomics, and metabolomics resources to gain insights into the metabolic functionalities of *R. irregularis* by reconstructing its high-quality genome-scale metabolic network that considers enzyme constraints. Extensive validation tests with the enzyme-constrained metabolic model demonstrated that it can be used to (i) accurately predict increased growth of *R. irregularis* on myristate with minimal medium; (ii) integrate enzyme abundances and carbon source concentrations that yield growth predictions with high and significant Spearman correlation ($\rho_s = 0.74$) to measured hyphal dry weight; and (iii) simulate growth rate increases with tighter association of this AMF with the host plant across three fungal structures. Based on the validated model and system-level analyses that integrate data from transcriptomics studies, we predicted that differences in flux distributions between intraradical mycelium and arbuscles are linked to changes in amino acid and cofactor biosynthesis. Therefore, our results demonstrated that the enzyme-constrained metabolic model can be employed to pinpoint mechanisms driving developmental and physiological responses of *R. irregularis* to different environmental cues. In conclusion, this model can serve as a template for other AMF and paves the way to identify metabolic engineering strategies to modulate fungal metabolic traits that directly affect plant performance.

**IMPORTANCE** Mounting evidence points to the benefits of the symbiotic interactions between the arbuscular mycorrhiza fungus *Rhizophagus irregularis* and crops; however, the molecular mechanisms underlying the physiological responses of this fungus to different host plants and environments remain largely unknown. We present a manually curated, enzyme-constrained, genome-scale metabolic model of *R. irregularis* that can accurately predict experimentally observed phenotypes. We show that this high-quality model provides an entry point into better understanding the metabolic and physiological responses of this fungus to changing environments due to the availability of different nutrients. The model can be used to design metabolic engineering strategies to tailor *R. irregularis* metabolism toward improving the performance of host plants.

**KEYWORDS** *Rhizophagus irregularis*, metabolic modeling

More than two-thirds of all land plants are involved in symbiotic relationships with arbuscular mycorrhizal fungi (AMF) (1). AMF are members of a monophyletic group within the early diverging fungi. Arbuscular mycorrhizal symbiosis is established by fungal hyphae entering cortical root cells of the host plant to form subcellular structures, termed arbuscles (ARB), where nutrients are exchanged between the symbiotic

Address correspondence to Zoran Nikoloski, nikoloski@mpimp-golm.mpg.de.

The authors declare no conflict of interest.

mSystems®

partners (2, 3). *Rhizophagus irregularis* (previously wrongly ascribed to *Glomus intraradices* [4]) is one of the most extensively studied AMF, shown to form symbioses with a variety of agriculturally relevant plants. Soil inoculation with *R. irregularis* leads to improved overall plant growth (5–8), fruit quality (9, 10), and yield (11–14). Further, *R. irregularis* confers robustness against multiple abiotic stress conditions (15–22). These qualities make it a valuable contributor to plant fitness, which is widely exploited for plant cultivation.

Spores of *R. irregularis* grow into a network of coenocytic hyphae, which can be separated into three major structures: the extraradical mycelium (ERM), the intraradical mycelium (IRM), and ARB (2). The ERM is comprised of hyphae located in soil, whereas hyphae of the two apoplastic structures, IRM and ARB, grow between or penetrate cortical root cells. *R. irregularis* mainly provides inorganic phosphate ($P_i$) and nitrogen (N) to the host plant as its extensive hypha network bridges the nutrient depletion zone surrounding the roots (23–27); in return, it receives carbohydrates and lipids from the host plant (28–35). $P_i$ is one of the key nutrients that limits plant growth, and under $P_i$-limiting conditions, most plants rely on additional $P_i$ supplied by a fungal symbiotic partner (3). To this end, the external hyphae of the fungus either take up $P_i$ directly from the soil or obtain it from hydrolysis of complex organic phosphates, such as phytate (36). According to the current evidence on *R. irregularis*, assimilated $P_i$ is polymerized into polyphosphate (PolyP), which is translocated through the ERM toward IRM (27). Finally, $P_i$ is released from arbuscles into the periarbuscular space. Several $P_i$ transporters have been identified in *R. irregularis* that could be involved in $P_i$ translocation from fungus to plant (26, 37, 38).

Moreover, N is another key nutrient for plant growth, comprising up to 5% of their dry weight. However, the availability of N sources to the plant is restricted due to the limited range of roots and its inhomogeneous distribution in soil. Hence, many plants depend on interactions with microbes that can provide additional nitrogen assimilated from the surrounding soil (39). *R. irregularis* takes up N in the form of ammonia ($NH_4^+$) and nitrate ($NO_3^-$) as well as amino acids and small peptides via designated transporters. Three $NH_4^+$ transporters, GintAMT1 to -3, and a $NO_3^-$ transporter, GiNT, have been identified in *R. irregularis* (40–43). Intracellular $NH_4^+$ is then used to synthesize L-arginine from L-glutamate (25, 43). Arginine is assumed to be the major transport form of nitrogen from the ERM to IRM, where it is catabolized to $NH_4^+$ and excreted into the periarbuscular space (3, 43).

The fungus, in turn, is dependent on carbohydrates and lipids obtained from the plant host. Multiple sugar transporters have been found that are likely involved in hexose transfer from the host plant to *R. irregularis* (31, 44). However, the sugars obtained from the plant are not sufficient for the fungus to complete its life cycle (i.e., formation of fertile spores). *R. irregularis* cannot synthesize fatty acids with chain length greater than eight due to the absence of the fatty acid synthase (FASI) and, thus, depends on fatty acids provided by the host plant (32, 33, 35, 45, 46). Most likely, lipid is transported as 2-monopalmitin; however, it has also been shown that *R. irregularis* can grow on myristate (47). These findings have been exploited to develop an axenic culture medium on which the obligate biotroph can grow up to the production of fertile spores (48).

The availability of an assembled genome for *R. irregularis* (49–52) largely facilitated the characterization of transporters and its lipid metabolism (45, 53), allowing us to draw conclusions about the metabolic capabilities of the obligate biotrophic fungus. Multiple studies performed gene expression profiling under various conditions, facilitating a deeper understanding of the *R. irregularis* metabolism and arbuscular mycorrhiza (5, 54–56). An annotated genome of an organism is also the basis for the generation of genome-scale metabolic models (GEMs) that offer the means to *in silico* probe the functional capabilities and physiological responses of the organism (57). GEMs have already been developed to analyze the interaction of an N-fixing bacterium, *Sinorhizobium meliloti*, and its host plant, *Medicago truncatula* (58, 59). As a result, important features of the N exchange and codependent growth were revealed, leading

to a better understanding of this symbiotic relationship. Such analyses for *R. irregularis* cannot be performed due to the lack of a high-quality GEM for this organism.

Availability of a GEM for *R. irregularis* can be particularly useful to dissect mechanisms underlying arbuscular mycorrhiza and to predict fungal nutrient conversions and exchange, directly affecting growth of the host plant. Here, we present a compartmentalized enzyme-constrained GEM for *R. irregularis*, termed iRi1574, which allows the integration and prediction of transcript and protein abundances for different growth scenarios. We then used the enzyme-constrained GEM of *R. irregularis* to predict protein abundances across four carbohydrate sources and three feeding concentrations; we also examined the predictions of growth and pathways that affect this complex phenotype based on experimental measurements of hyphal dry weight and protein content from Hildebrandt et al. (60). We show that the enzyme-constrained iRi1574 model results in predictions that correlate well with experimentally measured dry weight (as well as calculated growth rates) and allows us to probe the flux redistributions across three fungal structures using reanalyzed published gene expression data (5). Thus, we lay the foundation for further in-depth analysis of *R. irregularis* metabolism, hypothesis testing regarding mechanism essential for arbuscular mycorrhiza, and metabolic engineering of this fungus to improve the effect on agriculturally relevant plant traits.

## RESULTS AND DISCUSSION

**Reconstruction of the genome-scale metabolic model of *R. irregularis*.** Our first contribution is the generation of a GEM for *R. irregularis* encompassing all enzymatic functions annotated for this agronomically relevant fungus. The metabolic model can be used in combination with computational approaches from the constraint-based modeling framework to predict a variety of metabolic phenotypes, including growth, in different scenarios (61, 62). The genome of *R. irregularis* (49, 51) was used as a starting point for the generation of the GEM using the KBase fungal reconstruction pipeline (63). The resulting draft model was first translated to a common namespace based on augmenting a database of biochemical reactions, ModelSEED (34), since there were reaction and metabolite identifiers from published fungal models without cross references. We then added 198 transport reactions from the *Saccharomyces cerevisiae* iMM904 GEM (64) to improve the network connectivity (see Table S2H in the supplemental material). We further expanded the list of reactions based on literature evidence for *R. irregularis*. After these steps, the model was manually curated to ensure mass and charge balancing. Finally, stoichiometrically balanced cycles were removed from the model to avoid simulations in which growth without available carbon source is possible (Text S1).

The manually curated GEM of *R. irregularis*, named iRi1574, consists of 1,286 metabolites and 1,574 reactions in eight subcellular compartments, i.e., the cytosol, mitochondrion, peroxisome, Golgi apparatus, endoplasmic reticulum, nucleus, vacuole, and an extracellular compartment. In total, 687 enzyme-coding genes are associated with 1,054 (67%) reactions via gene-protein-reaction (GPR) rules (Fig. 1C). Further, we cross-referenced both metabolites and reactions to commonly used biochemical databases to increase the comparability to other GEMs and to facilitate its future use. A published cost-efficient medium that is used in dual-compartment culture systems and includes glycine, myoinositol, pyridoxine hydrochloride, thiamine hydrochloride, nicotinic acid, and essential minerals is the default medium for simulations (65). The dependence of the growth of *R. irregularis* on lipid transferred from the host (most likely 16:0 $\beta$-monoacylglycerol [32, 33]) was modeled by adding an exchange reaction for palmitate, which is added to the default medium.

Altogether, the iRi1574 model includes 13 metabolic subsystems (Fig. 1A). In total, 24% and 13% of reactions take part in lipid and amino acid metabolism subsystems, respectively, which dominate the reconstruction (Fig. 1A). To model the lipid metabolism of *R. irregularis*, we relied on the gene annotations supported by the literature (45, 66).

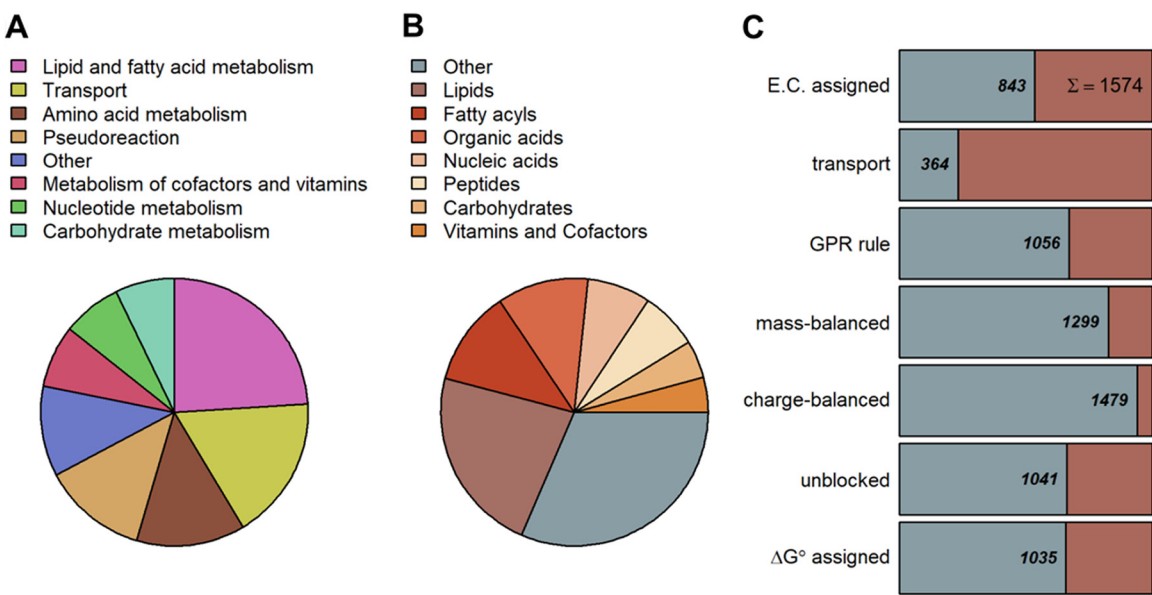

**FIG 1** Properties of the *R. irregularis* genome-scale metabolic model iRi1574. (A) The iRi1574 model includes 13 metabolic subsystems, primarily defined by KEGG pathways with manual refinement. The pie chart illustrates the percentage of reactions participating in these metabolic subsystems. (B) Metabolite classification using KEGG BRITE with manual refinement with help of the ChEBI ontology. (C) Binary classification of reactions based on eight criteria, including assignment of E.C. number, involvement in transport reactions, association with genes via GPR rules, mass and charge balancing, available value for standard Gibbs free energy, and ability to support steady-state flux.

Moreover, to incorporate experimentally measured lipid abundances (45, 67) into the biomass reaction, we used the SLIMEr method (68), whereby specific lipid species are split into their fatty acyl chains and backbone and are then combined using respective pseudoreactions. Hence, the number of lipid-related reactions and pseudoreactions is high compared to that of the remaining 11 metabolic subsystems. Based on evidence in the literature, we further added reactions that allow the production of ethylene (69), short-chain lipochitooligosaccharides (LCO) (70, 71), and vacuolar polyphosphate (72, 73). The respective end products of these reactions are exported via sink reactions. Moreover, we added extracellular sink reactions for organic phosphate and ammonia, as these molecules are known to be transported from the fungus to the host plant.

As only a small proportion of metabolites is annotated by the KEGG BRITE hierarchy (74), we used the ChEBI metabolite ontology (75) to structurally classify the considered metabolites. Due to the large number of reactions from lipid metabolism included in iRi1574, the proportions of lipids and fatty acyls are high (34%), followed by peptides/amino acids and organic and nucleic acids (Fig. 1B). The class of other metabolites is dominated by carbonyl compounds, heterocyclic compounds, and phosphosugars. Quality assessment tests with the iRi1574 model were performed by employing the MEMOTE test suite (76), yielding an overall score of 72% (Text S1).

**Comparison of iRi1574 to other fungal models.** As *R. irregularis* is phylogenetically distant from other fungi for which GEMs have been published, we next asked whether the phylogenetic relationship among these fungi is represented in the enzyme sets included in the respective GEMs. To this end, we assigned pathway information to the reaction in nine fungal models according to the classification contained in the YeastGEM v8.3.5 (77) model (Table S1A). To determine the overall similarity between two fungal models, we determined the overlap in E.C. numbers per subsystem by using the Jaccard index (JI). We observed that, compared to the nine compared fungal models, iRi1574 showed the lowest JI, i.e., lowest overlap of E.C. numbers, for fatty acid metabolism (including synthesis and elongation), thiamine metabolism, glycerolipid, and nicotinate and nicotinamide metabolism (Fig. S1). In contrast, the largest overlap was found for the pentose phosphate pathway, one carbon pool by folate,

pantothenate, and coenzyme A (CoA) biosynthesis, and amino sugar and nucleotide sugar metabolism, to name a few (Fig. S1). Further, we identified that some fungal GEMs show differences from iRi1574 with respect to particular metabolic subsystems. For instance, the model of *N. crassa* showed particular differences in the tricarboxylic acid (TCA) cycle and pyruvate metabolism, the model of *A. terreus* displays particular differences in purine metabolism, steroid biosynthesis, sphingolipid, and pyrimidine metabolism, and lipid metabolism, and the model of *P. chrysogenum* differed in sphingolipid metabolism and fatty acid elongation (Fig. S1).

The previous comparison between the fungal models was conducted only with respect to overlap of E.C. numbers present in particular metabolic subsystems and does not point at differences in the activity of these pathways and their contribution to the physiology of the modeled fungi. To address this issue, we employed flux balance analysis (FBA) (78, 79), which facilitates simulation of growth at steady state in each of the fungal models by optimizing of the flux, $v_{bio}$, through a biomass reaction that integrates the biomass precursors. This results in a linear optimization problem that imposes metabolic steady state and physiologically relevant bounds on reaction fluxes, i.e.,

$$\max v_{bio}$$

subject to (s.t.)

$$Sv = 0$$

$$v_i^{min} \leq v_i \leq v_i^{max}, \ \forall i \in R$$

where $S$ represents the stoichiometric matrix, including the molarity with which each substrate and product enter a reaction of the metabolic model, $v$ stands for the flux distribution, and $R$ denotes the set of reactions in the model. Since it is well-known that there are often multiple steady-state flux distributions, $v$, that achieve the same growth (80), to characterize the activity of a metabolic subsystem, we next determined the minimum and maximum values that the sum of fluxes of the reactions participating in a given subsystem attain at optimal growth (see Materials and Methods). Similarly, we determined the sums of fluxes from parsimonious FBA (pFBA) for each of the subsystems (Text S2).

Following this analysis, we observed that the ranges between the maximal and minimal sums of fluxes are largely overlapping and are of similar widths across most of the compared models (Fig. S2). Interestingly, the model for *P. chrysogenum*, iAL1006, and the iRi1574 model showed narrower ranges than the remaining models, except for fatty acid metabolism. The reason for this observation is the higher number of soft- and hard-coupled reactions to the biomass reaction in the iAL1006 and iRi1574 models compared to the others (Fig. S3). Moreover, we observed that the maximum sum of fluxes is similar across all fungal models (coefficient of variation [CV] of 0.6), while minimal sums and sums from pFBA fluxes showed larger differences (CV = 2.3 and CV = 2.6). This suggests that these pathways are of differential importance for the models, since the minimal sum of fluxes provides an indication of how much flux must at least pass through these reactions to guarantee optimal growth. In conclusion, we find differences in both E.C. number overlap as well as in the pathway activities between iRi1574 and other fungal models, indicating that iRi1574 is both structurally and functionally distinct from other fungal GEMs.

**iRi1574 can predict phenotypes of *R. irregularis*, in line with experimental observations.** We employed the assembled GEM to predict physiological traits for which there exist sufficient evidence and, thus, can be used to validate the performance of the model. A first question is how many of the reactions in the assembled model can carry flux. For these simulations, M-medium (65, 81) was used, which was enriched with palmitate, D-glucose, and D-fructose, assumed to be supplied by the plant (Table S1K). Using this default medium, 658 (42%) reactions were blocked (i.e.,

could not carry flux in any steady state supported by the model), of which 105 are transport reactions for extracellular metabolites. This is in line with the percentage of blocked reactions in the fungal models used in the comparison described above (from 11.9% in iJL1454 to 49.9% in iRL766).

An important characteristic of the symbiotic relationships formed by *R. irregularis* is its dependence on association with the plant host to ultimately form fertile spores (46). According to recent findings, lipids are supplied by the plant symbiont, as *R. irregularis* does not possess the required enzyme set for *de novo* synthesis of long-chain fatty acids from hexoses (35, 45). More specifically, 2-monopalmitate was proposed as a candidate for the lipid exchange from plant to fungus (32, 33). Concordantly, axenic growth of this fungus is only possible when fatty acids are supplied in the medium (47, 82). Hence, the default medium used in the study includes palmitate as a lipid source. Indeed, simulations in which palmitate influx is blocked lead to no growth with or without consideration of other carbon sources.

It has been shown that *R. irregularis* can utilize additional carbon sources (30, 47, 83). The ability of the model to reproduce these finding was assessed by growth simulations on single carbon sources in the default medium while restricting the uptake of palmitate to a minimal value that still guarantees optimal growth (8.46 mmol per gram dry weight [gDW$^{-1}$] h$^{-1}$). As a result, we simulated growth on 11 carbon sources by using FBA (described above), resulting in different growth rates (Fig. S4A). Here, we observed the highest growth rates for trehalose, followed by D-glucose, D-fructose, melibiose, and raffinose. The observed high growth rate with trehalose as a carbon source is not surprising, given that it directly enters the biomass reaction. The equal growth rates obtained upon adding D-glucose, D-fructose, raffinose, and melibiose indicated that D-glucose, D-fructose, and D-galactose as breakdown products from raffinose can be used with equal efficiency. The efficiency of the remaining carbon sources differed due to the differences in their breakdown pathway and additional modifications (e.g., phosphorylation and reduction). Further, we quantified the ATP production for each of these carbon sources under the same conditions while guaranteeing 0% or 50% of the optimal predicted growth (Fig. S4B and C). As a result, we observed that the highest ATP production can be achieved from utilizing trehalose, which is likely because it is directly used by the biomass reaction. Since the trehalose uptake is not growth-limiting, the excess trehalose can be used for ATP production. The highest ATP yield is achieved by myristate utilization, which confirms an experimental observation made by Sugiura et al. (47) that the ATP content increased by 2.4-fold in the presence of myristate.

Moreover, it has been reported that the addition of myristate to the medium leads to enhanced growth of *R. irregularis* (47). We found that optimum growth is, as expected, associated with a fixed value of palmitate influx of 8.46 mmol gDW$^{-1}$ h$^{-1}$. Further, myristate is not utilized if additional carbon sources are available without limit in the medium, which is in contrast to the experimental findings of Sugiura and co-workers, who found that the addition of myristate leads to an increment in growth irrespective of an additional carbon source (47). Therefore, we asked if the reduced growth, due to the suboptimal scenario of fixing the palmitate influx to 10% of the minimum at optimal growth, can be complemented by adding myristate. Indeed, the model predicted that growth increased by 1.5% compared to the suboptimal scenario. Notably, this growth complementation does not occur due to compensation of fatty acid production but via increased energy generation from products of $\beta$-oxidation. When additional carbon sources (i.e., D-glucose, D-fructose, glycine, and myo-inositol) are allowed, with uptake rates restricted to their minimal fluxes at optimal growth, this increase in growth amounts to 9.7% (see below for the predictions from the enzyme-constrained model).

Another important transport process described for this symbiosis is the transport of P$_i$ from fungus to the host plant (38). We found that the reconstructed model predicts export of P$_i$ at optimal growth (Table S1B for FVA), in line with available evidence (38).

These results corroborate the quality of the functionally relevant predictions based on the developed iRi1574 model.

**Protein usage with different carbon sources.** Enzyme-constrained GEMs have been developed for *S. cerevisiae* and *Escherichia coli* (68, 84, 85), demonstrating improved prediction of metabolic phenotypes, in contrast to the classical FBA-based models. In enzyme-constrained GEMs, the fluxes of reactions are bounded by the catalytic efficiency ($k_{cat}$ parameters) and the abundance of the respective enzyme(s) (86); these models also include constraints on the total enzyme content, borrowing from the initial idea proposed in FBA with molecular crowding (87, 88). An enzyme-constrained GEM can be used to predict not only growth but also distribution of the total enzyme content across the different reactions and pathways. To generate an enzyme-constrained GEM for *R. irregularis*, we made use of 1,214 $k_{cat}$ parameters, of which 430 were measured for fungi, covering 57.4% of reactions included in the model (with all irreversible reactions; see Materials and Methods). We then employed an extension of MOMENT (84), a constraint-based approach that facilitates the integration and prediction of protein abundance by considering data on the $k_{cat}$ values. In addition to a molecular crowding constraint (see equation 5 and Materials and Methods) (84, 87, 88), similar to GECKO (68), we introduced a constraint to model enzyme promiscuity (equation 4), resulting in the extended method we refer to as eMOMENT. Missing turnover numbers were accounted for by assigning the median of the assigned $k_{cat}$ values.

Here, we first revisit the results based on FBA with respect to growth on myristate and export of $P_i$. Without additional constraints in the enzyme-constrained iRi1574 model, the positive effect of myristate uptake on growth could not be reproduced, since myristate is catabolized via peroxisomal $\beta$-oxidation and the expression of the required enzymes is not outweighed by the benefit of generating acetyl-CoA from myristate. However, when the allocation of total protein is shifted from the optimal ratio toward increased abundances of peroxisomal proteins (Text S2), the addition of myristate can increase growth compared to the suboptimal scenario (Fig. S5). Further, we found that using the default medium, like in the FBA model described above, the enzyme-constrained model predicts export of $P_i$ at optimal growth in the range of 0 to 171.9 mmol gDW$^{-1}$ h$^{-1}$ (Table S1C). Therefore, the observations made for the FBA model with these important phenotypes also hold for the enzyme-constrained model.

To test the performance of the enzyme-constrained variant of the iRi1574 model, we made use of published dry weight and protein content available for 12 combinations of four carbon sources (i.e., D-glucose, D-fructose, raffinose, and melibiose) at three different concentrations (i.e., 10 mM, 100 mM, and 1,000 M) (60). These data were generated by using the *G. intraradices* strain Sy167 (60), which is the closest species to *R. irregularis* for which these kinds of measurements are available. The different medium conditions were modeled by adding each carbohydrate to the default medium as a single carbon source, while the respective concentrations were modeled as proportional uptake fluxes considering kinetic parameters of the respective transport reactions (for more details, see Materials and Methods). Like in the findings based on FBA described above, palmitate was present in the default medium since growth without palmitate is not possible, irrespective of additional supply of carbohydrates (45, 47). To avoid compensation of lower carbohydrate supply by $\beta$-oxidation of palmitic acid, we limited its uptake to the flux value obtained at optimal growth predicted by FBA.

We next compared the predictions of growth from the eMOMENT approach with those from FBA (i.e., without considering enzyme constraints), with the same restrictions on palmitate uptake (Table S1D). We observed that the additional constraints on protein abundances largely improved the quality of the prediction (Fig. 2) and resulted in values of the same order as growth rates calculated from dry weights and grow duration (Text S2). We found that the predicted growth rates were significantly correlated with the measured values for hyphae dry weight (Spearman correlation coefficient, $\rho_S = 0.74, P < 0.01$; Fig. 2) and were colinear ($\rho_S = 1.0$) with the protein content. In contrast, FBA predicted a statistically significant, negative correlation ($\rho_S = -0.69, P < 0.05$), demonstrating that the

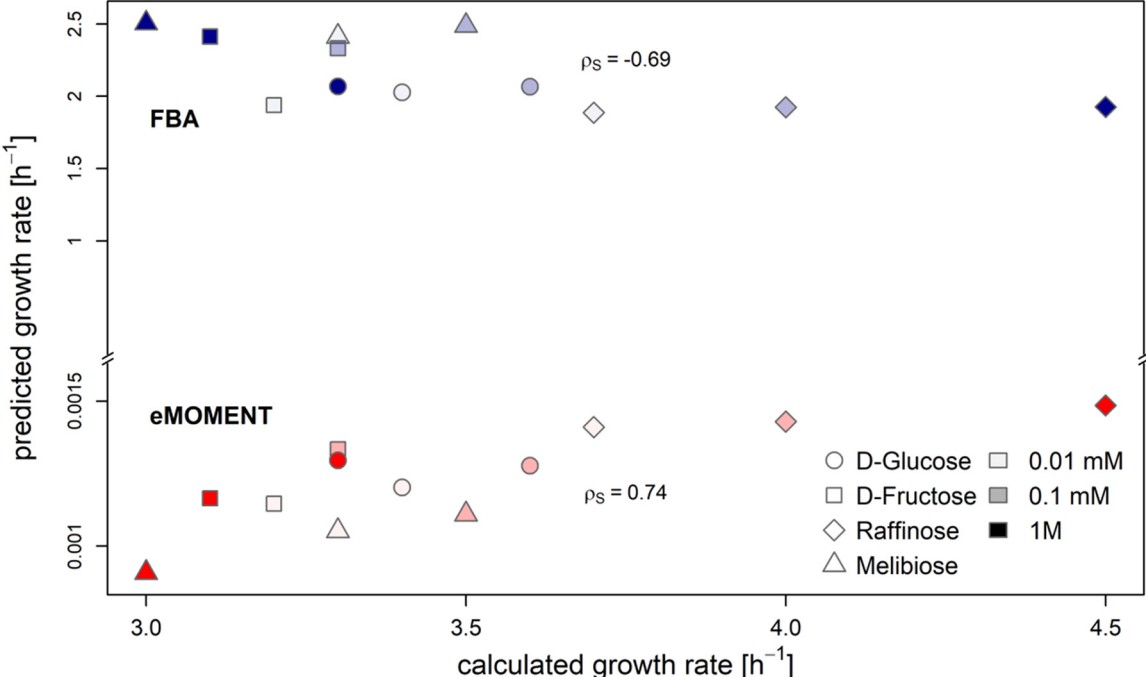

**FIG 2** Prediction of growth for iRi1574 using eMOMENT and FBA. Scatterplot of growth rates predicted by eMOMENT (red) compared with FBA without constraints on enzyme abundances (blue). The predicted growth rates were compared with experimental data obtained for *Glomus intraradices* Sy167 (60), which is the phylogenetically closest species with this kind of data available. The concordance of predicted growth rates and experimentally measured hypha dry weight was quantified by the Spearman correlation, $\rho_S$.

predictions from this approach are not in line with the experimental observations. The respective values for Pearson correlation were $\rho_P = 0.80$ ($P < 0.01$) for the enzyme-constrained models and $\rho_P = -0.62$ ($P < 0.05$) for the FBA model. Using FBA, we observed that growth increased with the concentration of the respective carbon source despite rescaling of biomass coefficients, while this was not the case when using the eMOMENT approach. In fact, this relationship was only observed for D-glucose and raffinose, which is broken down to sucrose and D-galactose extracellularly. Hence, the iRi1574 model can reliably predict growth based on different carbon sources when protein content and protein-reaction associations are considered. A reason for the significant negative correlation with FBA could be the altered biomass composition after rescaling by the changing protein content as well as the lack of protein constraints, which are present in the enzyme-constrained model.

The applied approach to integrate total protein content allowed us to predict not only optimal growth but also abundances of individual proteins for the 12 combinations of carbon source and concentrations considered. Since multiple allocations of proteins to enzyme complexes and reactions can lead to optimal growth, we sampled the set of feasible enzyme abundances (see Materials and Methods) at 99% of the respective optima. The resulting predictions on alternative enzyme allocation at optimal growth were used to investigate the plasticity of enzyme allocation under the different conditions. We quantified the plasticity in the abundance of each protein by the CV across the simulated conditions. The CV was calculated for predicted protein abundance and reaction flux across the 12 growth scenarios (Tables S1E and F). To illustrate the findings, we represented the distribution of CVs across the 13 metabolic subsystems (Fig. 3A and B). The highest median CV was found for enzymes within the amino sugar and nucleotide sugar metabolism (CV = 16.65), carbohydrate metabolism (CV = 9.71), and nucleotide metabolism (CV = 9.35). In contrast, metabolism of cofactors and vitamins and transport reactions showed the lowest plasticity in protein abundance (CV < 0.3). Regarding reaction fluxes, the subsystems containing the most plastic

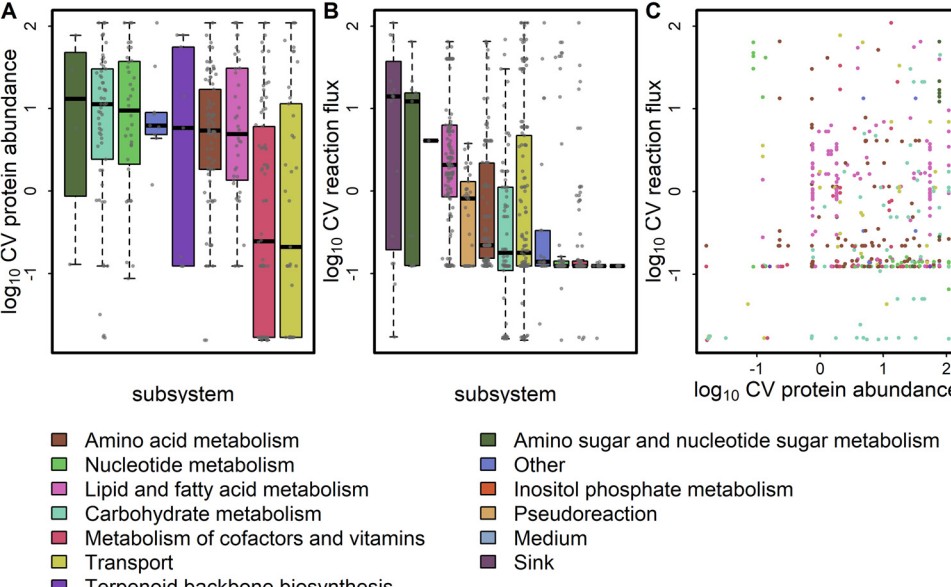

**FIG 3** Plasticity of protein abundance and reaction fluxes across 12 simulated media conditions. The coefficient of variation (CV) was calculated across all medium conditions (i.e., glucose, fructose, raffinose, and melibiose at 10, 100, and 1,000 mM each) for protein abundances (A) and reaction fluxes (B). The boxes are ordered by median of the log$_{10}$-transformed data. (C) The CV of fluxes is plotted against the CV of abundance of their associated proteins.

reactions were found within sink reactions (CV = 13.77) and amino sugar and nucleotide sugar metabolism (CV = 12.55).

To assess whether the plasticity in flux is dependent on the variability in enzyme abundance of the catalyzing enzymes, we compared the respective sets at the extreme ends of CV distribution (10% and 90% quantiles) between protein abundance and reaction fluxes (Fig. 3C). Among the 89 reactions associated with enzymes with variable abundance (CV ≥ 49.38), 28 also were found to be highly plastic in flux. Conversely, we found six reactions with low flux CV associated with seven high-abundance CV proteins. Four of these genes were not promiscuous and were associated with single reactions of high CV. The associated reactions are involved in terpenoid backbone biosynthesis and nucleotide metabolism. Hence, variation in enzyme abundance cannot fully explain the plasticity in flux. Since pH differences (affecting enzyme activity) are expected to lead to systemic changes, we conclude that the plasticity in flux for these selected reactions is largely driven by metabolite concentration rather than enzyme abundance.

Among the 78 reactions with highly variable fluxes (CV ≥ 39.79), the majority lie within the lipid and fatty acid metabolism (47) and transport reactions (10). The subset of reactions in lipid metabolism was found to act mainly in in lipid degradation but also in the synthesis of very long-chain fatty acids. This result indicates a trade-off between lipid synthesis and β-oxidation depending on the type and concentration of the carbon source.

**Prediction of growth for three fungal structures.** As obligate biotrophs, AMF are dependent on the association with a host plant for carbohydrates and lipids (2, 3). Three major fungal structures are discriminated for the fungus: extraradical mycelium (ERM), intraradical mycelium (IRM), and arbuscles (ARB), which differ from each other in the proximity of association with the host plant. Thus, we investigated growth and underlying flux distributions comparing these three structures of *R. irregularis*. To this end, we used published expression data (5) to examine growth and differential reaction fluxes between these three structures. We relied on using transcript abundances instead of protein abundances, as the latter had not been measured at the time this

mSystems®

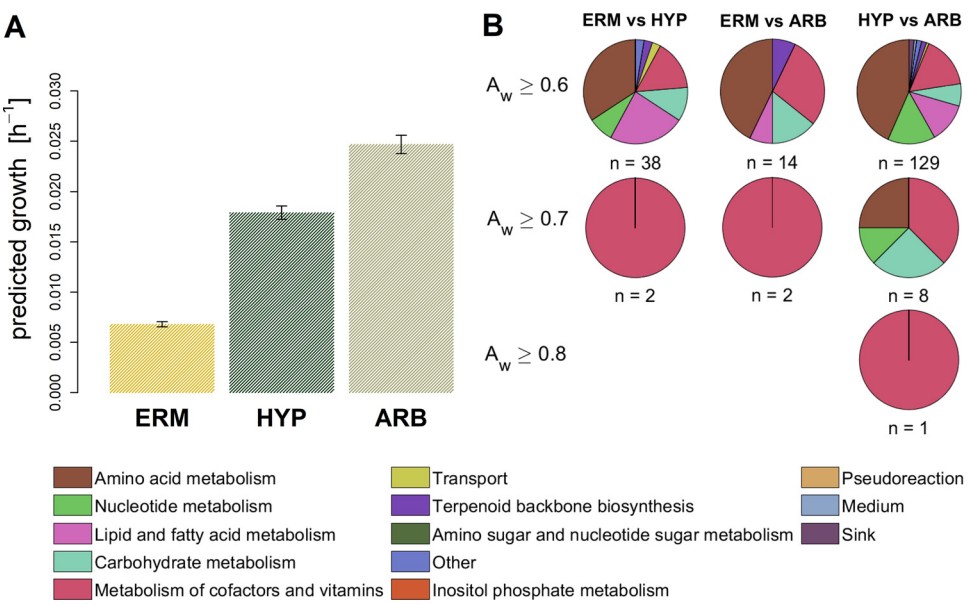

**FIG 4** Growth simulation of *R. irregularis* for three fungal structures. The upper limit for reaction flux was calculated as $k_{\text{cat}} \cdot [E]$. The association of turnover numbers with reactions was done similar to GECKO (68). Structure-specific expression data were used as proxy for protein concentrations. This was done by multiplying relative transcript abundances with the maximum total protein content measured with the available carbon source ($C = 0.106$ g gDW$^{-1}$) (60) (see Materials and Methods for more details). (A) Predicted growth for the three fungal structures. The error bars represent predicted growth rates at $C \pm \sigma$, where $\sigma$ represents the standard deviation determined for the experimentally measured protein content. (B) Distribution of subsystems for reactions that show nonparametric common language effect sizes ($A_w$) above selected thresholds for each pairwise comparison of flux distributions between the three fungal structures. The total numbers of reactions with $A_w$ greater than the threshold are shown below each of the pie charts. No chart is shown if no reaction was found to have an $A_w$ above the threshold.

study was conducted. However, transcript abundances have been successfully used to constrain fluxes in other constraint-based metabolic modeling studies (89, 90).

We observed an increase in growth upon association of the fungus with the host plant (Fig. 4A), which was expected since a tighter association with the host plant and, hence, increased nutrient uptake allows faster growth. Since the total protein content remained the same over the simulations for all three structures, the changes in growth likely result from increased flux through a subset of reactions responsible for growth due to the larger upper bounds of these reactions. One reason for this could be changes in the relative abundances of individual proteins due to changes in transcript abundances that were used to calculate the upper bounds. To determine differential reactions, we sampled 5,000 flux distributions for each structure and compared the resulting flux values for each reaction using the nonparametric common-language effect size ($A_w$) (91) (Table S1G). We used three different thresholds for $A_w$ (i.e., 0.6, 0.7, 0.8) to find differentially activated reactions between each pair of structures. By using 0.6 as a threshold, we found that mainly reactions of the amino acid metabolism exhibited differential fluxes between each pair of structures, followed by reactions in metabolism of cofactors and vitamins, carbohydrate, lipid, and nucleotide metabolism (Fig. 4B). Upon increasing the threshold to 0.7, we found only two reactions to be differentially activated between ERM versus IRM and ARB, which were both involved in riboflavin biosynthesis (KEGG entry M00911) (Fig. 4B). Moreover, eight reactions from metabolism of cofactors and vitamins were differential between IRM and ARB. When the threshold was increased to 0.8, only one reaction was found to differ between IRM and ARB, namely, the coproporphyrinogen-oxygen oxidoreductase (E.C. 1.3.3.3). These results suggest that substantial rerouting of fluxes within these pathways occurs upon establishing the fungus-plant interface. However, differences in predicted growth may

not exclusively result from large changes in a few reactions. It is likely that small changes in a number of other reactions also contribute to an increased growth rate.

**Conclusions.** Although *R. irregularis* is one of the most extensively studied AMF that forms symbioses with major crops, insights from the annotation of its enzymatic genes, the extensive body of evidence about its physiological and molecular responses to different environmental stimuli, and mutual effects on plants with which it interacts have not yet been systematically investigated in the context of metabolic modeling. The constraint-based modeling framework allows us to dissect the molecular mechanisms that underpin these responses and also to suggest targets for future metabolic engineering to boost the beneficial effects of this AMF. However, achieving this aim requires the assembly of a high-quality large-scale model that leads to accurate quantitative predictions of multiple traits in different scenarios.

Here, we presented the enzyme-constrained iRi1574 GEM of *R. irregularis* based on the KBase fungal reconstruction pipeline followed by consideration and inclusion of exhaustive literature research as well as manual curation for consistency and mass and charge balance. One possible caveat of using fungal reconstruction pipelines is that the resulting model may be very similar to the employed templates. Nevertheless, by conducting comparative analyses of the enzyme set of iRi1574 and that of published fungal models, we demonstrated the specificity of iRi1574 and its ability to capture the particularities of *R. irregularis* metabolism. More importantly, validation tests demonstrated that iRi1574 can (i) accurately predict increased growth on myristate with minimal medium with the FBA model as well as under additional constrains on enzyme distribution in the enzyme-constrained model, (ii) predict growth that is highly correlated with hyphal dry weight measured in a close relative (*Glomus intraradices* Sy167, neighboring clade) when considering enzyme constraints, and (iii) increase growth rate with tighter association with the host plant based on integration of relative transcriptomics data. The extensively validated model was used to show that the transition from IRM to ARB could be linked with changes in amino acid and cofactor biosynthesis.

This first model of an AMF can be coupled with root-specific models of model plants to investigate the effects of symbiosis. Further, a two-dimensional (2D) growth simulation approach (62) can be employed to obtain a realistic growth measure for hyphal spread. In addition, the iRi1574 model can be used to mechanistically dissect the interactions of species in fungal and bacterial communities that jointly affect plant performance (92). Most importantly, one can begin to design metabolic engineering strategies to improve desired traits in *R. irregularis*, study the effect of the modifications on plant performance by coupling metabolic models of the symbionts, and further refine the model based on integration of heterogeneous molecular data. Altogether, these modeling efforts can guide future reverse genetics tools used to understand the functional relevance of metabolic genes in *R. irregularis* in shaping plant traits.

## MATERIALS AND METHODS

**Draft model generation.** The genome of *Rhizophagus irregularis* DAOM 181602 (DAOM 197198; GCF_000439145.3) (49, 51) served as the basis for the genome-scale metabolic reconstruction. The initial draft model was obtained from KBase (63) using the Build Fungal Model app (15 October 2018; narrative ID 36938). The resulting model was gap-filled with the help of the KBase Gapfill Metabolic Model app using complete medium. A set of 35 additional reactions was required to simulate growth. This set of added reactions was manually curated in the next step of model refinement. The gap-filled model was then downloaded in SBML format and further modified within MATLAB (93) using functions of the COBRA toolbox (94).

**Model curation.** To enhance connectivity between the cellular compartments, 198 transport reactions were added from the yeast iMM904 metabolic model (64). The imported transport reactions were validated during the next curation steps. Out of all added transport reactions, 71 were kept in the model despite missing literature evidence (see Table S1H in the supplemental material). Next, the metabolite and reaction identifiers were translated, whenever possible, to the ModelSEED namespace (34). This step was necessary since the identifiers resulted from 14 different models and the catalyzed reactions mostly could not be identified. Moreover, this led to a higher connectivity of the network as identical metabolites and reactions were reconciled. Further, cross-references were added to BiGG (95), MetaCyc (96), KEGG (74), MetaNetX (97), PubChem (98), and E.C. numbers.

Metabolite formulas were added from PubChem and adapted to the net charge at the average

cytosolic pH of 6.2 (99) using ChemAxon Marvin software (Marvin 17.21.0, 2017, http://www.chemaxon .com). With elemental compositions and metabolite charges available, the model was manually mass and charge balanced.

After these steps, additional reactions were added from various literature sources. Most of the lipid metabolism is based on the results from reference 45, including sterol metabolism, fatty acid synthesis, elongation and degradation, glycerolipid metabolism, and sphingolipid metabolism. Plasma membrane transporters were added with literature evidence from multiple sources (38, 44, 53, 100). Furthermore, important dead-end metabolites were resolved manually by adding incident reactions with genomic evidence or transport reactions.

The biomass reaction was adapted from the default fungal biomass reaction added during the automated reconstruction process (Table S1I). Subsequently, the unknown coefficients in the biomass reaction were rescaled such that the sum of coefficients multiplied with the respective molecular weight equals 1 g gDW$^{-1}$ (101). Due to missing experimental data, we set the growth-associated ATP maintenance reaction (GAM) to 60 molecules ATP gDW$^{-1}$ as taken from the KBase default fungal biomass, which is in line with the average value from seven published fungal models (68.87 mmol gDW$^{-1}$; Table S1J). The non-growth-associated ATP maintenance reaction (NGAM) was fixed to the average of from seven published fungal models (3.21 mmol gDW$^{-1}$ h$^{-1}$; Table S1J). For the lipid component in the biomass reactions, the SLIMEr formalism was used (102), and coefficients of tail and backbone pseudo-metabolites were adjusted to render the model feasible for simulations by running a quadratic program to minimize factors to be added to the respective coefficients.

Stoichiometrically balanced cycles (SBC) were then removed by repeatedly applying flux variability analysis (FVA) and correcting reaction reversibility or adding additional reactions as suggested previously (103). For the following analyses, all reversible reactions were split into two irreversible reactions.

**Short-chain CO and LCO.** Synthesis reactions for LCOs were added by first modeling the synthesis of COs with chain lengths of 3 to 6 with subsequent acetylation reactions adding 16:0, 16:1Δ9($\omega$7), 18:0, and 18:1Δ9($\omega$9) fatty acids, leading to 16 different LCO species (70, 71).

**Transcriptomic data.** Structure-dependent transcriptome sequencing (RNA-seq) data were obtained as raw sequence reads (GenBank accession no. GSE99655) (5). The reads were quality trimmed using Trimmomatic-0.39 (104) and mapped to the *R. irregularis* genome using STAR 2.7.3a (105). The read quantification was performed using HTSeq count (106). The average over the three replicates was used for further analysis. The protein identifiers from the original study were translated to the identifiers of the genome annotation that was used for the metabolic reconstruction using local tblastn (107, 108) with the BLOSUM90 scoring matrix and a cutoff E value of 10E−90. The average Spearman correlation between the published and reanalyzed values for the secreted proteins (SP) was 0.8, which confirms the previous results given different analysis software and possible mapping errors using tblastn.

**Turnover numbers.** For the assignment of $k_{cat}$ values to reactions, an approach similar to that for GECKO (68) was applied. First, turnover values for all E.C. numbers in the model across all organisms and lineages were obtained from BRENDA (109), SABIO-RK (110), and UniProt (111). For each E.C. number assigned to a reaction, all matching $k_{cat}$ values were obtained and, if possible, filtered for substrate matches and enzymes from the fungi kingdom. If no match for the complete E.C. number was found, the same procedure was applied to the same E.C. number pruned to a lower level. Among the obtained values, the maximum $k_{cat}$ value was assigned to the respective reaction. The distribution and numbers of matched $k_{cat}$ values per subsystem, as well as a comparison to $k_{cat}$ values in the YestGEM v8.3.4, are shown in Fig. S6A and B. The median of all nonzero values was used for metabolic reactions without a matched $k_{cat}$ value. To arrive at units of per hour, all turnover numbers were multiplied by 3,600.

**Enzyme use under different growth conditions.** To predict the enzyme abundances with different medium conditions, four different carbon sources (i.e., D-glucose, D-fructose, raffinose, and melibiose) were added to the minimal medium (65) (Table S1K) as single carbon sources. These carbohydrates were selected as hyphal weight and protein content were available for them at three different concentrations (i.e., 10, 100, and 1,000 mM) (60). As an exception, palmitate was retained in the medium, as it must be supplied to the fungus to allow for growth (45, 47). We used kinetic parameters (i.e., $V_{max}$ and $K_m$) of *S. cerevisiae* monosaccharide transporters to model the influx of D-glucose, D-fructose, and D-galactose (results from breakdown of both raffinose and melibiose are in Table S1L) (112, 113). The respective upper bound for the transporters was calculated as

$$v = \frac{V_{max} \cdot [S]}{K_m + [S]}. \qquad (1)$$

Further, the import of palmitate was restricted to the flux value of 8.46 mmol gDW$^{-1}$ h$^{-1}$ at optimal growth as predicted by FBA.

The following MILP, which we termed eMOMENT, imposes constraints that were adopted from the MOMENT approach (84), which were extended by an additional constraint (equation 4):

$$\max v_{bio}$$

s.t.

$$Sv = 0 \qquad (2)$$

$$0 \leq v_i \leq E_i^r \cdot k_{\text{cat}i}^{\max}, \ \forall i \in R \tag{3}$$

$$\sum_{i \in \text{GPR}_k} E_{k,i}^r = E_k^g, \ \forall k \in G \tag{4}$$

$$\sum_k E_k^g \cdot \text{MW}_k \leq C, \ \forall k \in G \tag{5}$$

$$\alpha \cdot y_k \leq E_k^g \leq \beta \cdot y_k, \ \forall k \in G \tag{6}$$

$$y_k \in \{0,1\} \tag{7}$$

$$\alpha = 10^{-10} \text{ mmol gDW}^{-1}, \ \beta = 1 \text{ mmol gDW}^{-1} \tag{8}$$

where $R$ and $G$ represent the sets of reactions and genes and constraints on $E_j^r$ are imposed by the GPR rules. The molecular weight in grams per millimole protein, $k$, is given by $\text{MW}_k$. The constraint in equation 3 imposes an upper limit on the flux through the reaction, $i$, which is the product of the reaction-specific turnover rate and the enzyme abundance, $E_i^r$, available for this reaction. Further, binary variables, $y$, were introduced to indicate that the respective genes are expressed ($y = 1$) or not ($y = 0$). This was done to enforce a lower bound, $\alpha$, for the abundance of expressed genes to avoid numerical problems. The value for $E_i^r$ is determined by the GPR rules. To model the GPR rules, the following constraints were applied recursively in case of complex rules: (i) A AND B $\rightarrow E_i^r = \min(E_A^g, E_B^g)$, $E_i^r \leq E_A^g$, $E_i^r \leq E_B^g$; (ii) OR B $\rightarrow E_i^r = E_A^g + E_B^g$, $E_i^r \leq E_A^g + E_B^g$.

Further, the total protein content, $C$, was determined by the experimentally measured protein contents at the given concentrations (60). To account for changing protein contents, the coefficients of the biomass reaction were rescaled to the respective values for $C$. The proportions of the remaining biomass components were conserved when they were adapted to the new residual mass fraction (1 g mmol gDW$^{-1}$ − $C$).

We extended the constraints we borrowed from the MOMENT approach by one additional constraint (equation 4), which takes the promiscuity of proteins for multiple reactions into account. Hence, the abundance of protein $k$ is smaller than or equal to the sum of enzyme abundances across all reactions with which it is associated.

The feasible abundance ranges for all proteins were determined by individual minimization and maximization for $E_i^g$ at optimal growth, similar to FVA. Using these, we sampled 1,000 abundances compatible with the constraints above, by finding the closest vector of abundances to a randomly created set of abundances, $E^{g*}$, within the feasible ranges determined in the step before

$$\min |E^{g*} - E^g|$$

s.t.

$$v_{\text{bio}} \geq 0.99 \cdot v_{\text{bio}}^{\text{opt}} \tag{9}$$

where constraints are equations 2 to 8.

**Metabolic changes between fungal structures.** For the metabolic change experiment, all four carbon sources that were used in the analysis described above were added to the same minimal medium. Similarly, the upper bounds on monosaccharide import were calculated using transporter kinetics from *S. cerevisiae*, considering only the maximum concentration of 1 M. Across the calculated values, the maximum possible influx for each monosaccharide was selected. For this experiment, palmitate was also retained in the medium with the same upper limit as that described above. For each of the three structures (ERM, IRM, and ARB), the abundance of each protein, $\text{tc}_i^p$, was calculated from the relative transcriptomic counts per gene, $\text{tc}_i^g$ (not considering alternative splicing and posttranslational modifications):

$$\text{tc}_k^{g'} = \frac{\text{tc}_k^g}{\sum_k \text{tc}^g}, \tag{10}$$

$$\text{tc}_k^p = \frac{\text{tc}_k^{g'} \cdot C}{\text{MW}_k}. \tag{11}$$

The total protein content was set to the maximum value measured across all growth conditions used in the experiment before ($C = 0.106$ g gDW$^{-1}$). By applying this transformation, we assume that transcript levels correlate with protein abundances, which is not necessarily true and can lead to over- or underestimation of protein levels. However, this represents the closest approximation of protein levels in the absence of quantitative proteomics data.

To conduct FBA, the transformed transcript count, $\text{tc}_i^r$, for the reaction was first calculated by applying the GPR rules taking the minimum $\text{tc}^p$ value for complexes (AND) and the maximum for isozymes

(OR). Finally, the upper limit for a reaction, *i*, was defined as the product of estimated enzyme abundance and the respective turnover value:

$$v_i \leq k_{\text{cat},i} \cdot \text{tc}_i^r. \tag{12}$$

Growth was predicted for each of the three structures by FBA using the adapted reaction limits. After this, FVA was used to determine the feasible ranges for each reaction while keeping the growth at 99% of the optimum. These ranges were used as the limits for the sampling procedure which attempts to find an optimal solution with minimal distance to a random flux vector $v^*$:

$$min \ |v^* - v|$$

s.t.

$$Sv = 0$$

$$v_i \leq k_{cat,i} \cdot \text{tc}_i^r, \forall j \in R\#, \ v_{\text{bio}} \geq v_{\text{bio}}^{\text{opt}} \tag{13}$$

Like this, 5,000 points were sampled and used for a reaction-wise comparison between the three structures. To this end, the nonparametric estimate for common language, $A_w$ (91), was used to determine substantial changes of reaction flux between each pair of structures:

$$A_w = \frac{\left( \#(p > q) + 0.5 \cdot (p = q) \right)}{(n_1 \cdot n_2)}. \tag{14}$$

The variables *p* and *q* represent the vectors of sampled fluxes for the same reaction at two different structures.

**Data availability.** All procedures, data, and approaches used are available at https://github.com/pwendering/RhiirGEM.

## SUPPLEMENTAL MATERIAL

Supplemental material is available online only.

**TEXT S1**, DOCX file, 0.03 MB.
**TEXT S2**, DOCX file, 0.03 MB.
**FIG S1**, TIF file, 2.8 MB.
**FIG S2**, TIF file, 2.2 MB.
**FIG S3**, TIF file, 2.5 MB.
**FIG S4**, TIF file, 1.5 MB.
**FIG S5**, TIF file, 1.6 MB.
**FIG S6**, TIF file, 1.8 MB.
**TABLE S1**, XLSX file, 0.3 MB.

## ACKNOWLEDGMENTS

Z.N. acknowledges support from the Max Planck Society. Z.N. and P.W. acknowledge the funding from the Research Focus "Evolutionary Systems Biology" of the University of Potsdam.

P.W., designed and performed research and wrote the paper; Z.N. designed research and wrote the paper.

We have no competing interest to declare.

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
