## [Reviewer comments · mSystems]

Genome-scale modeling specifies the metabolic capabilities of *Rhizophagus irregularis*

Philipp Wendering and Zoran Nikoloski

Corresponding Author(s): Zoran Nikoloski, University of Potsdam

Review Timeline:

Submission Date:	October 7, 2021
Editorial Decision:	November 26, 2021
Revision Received:	December 23, 2021
Accepted:	January 3, 2022

Editor: Laura Hug

Reviewer(s): Disclosure of reviewer identity is with reference to reviewer comments included in decision letter(s). The following individuals involved in review of your submission have agreed to reveal their identity: Junling Shi (Reviewer #1)

Transaction Report:

DOI: <https://doi.org/10.1128/mSystems.01216-21>

November 26, 2021

Prof. Zoran Nikoloski
University of Potsdam
Bioinformatics, Institute of Biochemistry and Biology
Karl-Liebknecht-Str. 24-25
Potsdam 14476
Germany

Re: mSystems01216-21 (Genome-scale modeling specifies the metabolic capabilities of *Rhizophagus irregularis*)

Dear Prof. Zoran Nikoloski:

Thank you for submitting your manuscript to mSystems. We have completed our review and I am pleased to inform you that, in principle, we expect to accept it for publication in mSystems. However, acceptance will not be final until you have adequately addressed the reviewer comments. In particular, please respond to Reviewer #2's queries. Reviewer #1 notes that your model validation is based on cited studies rather than your own observations of organism growth - please make this more clear in the main text. Additional growth experiments are not required as a component of this revision.

Preparing Revision Guidelines

Sincerely,

Laura Hug

Editor, mSystems

Journals Department
Reviewer comments:

Reviewer #1 (Comments for the Author):

I noted the experimental data of the fungal biomass was cited from other reports. This should be done by the authors by themselves, since the model was constructed on the assumption basis that was made under certain conditions. Therefore, experimental data are needed to verify the model.

Reviewer #2 (Comments for the Author):

This manuscript offers a new genome-scale model of *R. irregularis*. The model is validated with significant experimental data, which demonstrates the value of the model. The model is shown to be somewhat quantitatively predictive of experimental phenotypes. Overall, the manuscript is well written with a novel model and some novel methods which should be of general interest. I am left with only a few questions and concerns, although in full disclosure, I have reviewed this manuscript previously and the authors have already responded to my initial round of questions and concerns. Here I am reviewing the manuscript as it stands now in the context of submission to msystems.

Questions and comments:

- 1.) Any thoughts as to why the fluxes from iAL1006 and the iRi1574 had a narrower range compared to other models? The result is mentioned around line 218 but never really explained.
- 2.) Can the authors speak to the energy metabolism of their model (a critical detail in all models)? What ATP yield is produced from the input sugars?
- 3.) The myristate results described around line 265 are somewhat confusing. The implication seems to be that myristate can replace a portion of the palmitate for fatty acid production? Is this correct?
- 4.) Can the authors briefly explain in the manuscript why eMOMENT products strong correlations while FBA produces negative correlations with experimental data?
- 5.) In the study described around line 369, was transcriptomic data used instead of protein abundance data in the model, and can the authors justify that in these conditions the two datatypes are interchangeable?

Reviewer #1 (Comments for the Author):

I noted the experimental data of the fungal biomass was cited from other reports. This should be done by the authors by themselves, since the model was constructed on the assumption basis that was made under certain conditions. Therefore, experimental data are needed to verify the model.

We have used all available experimental data for *R. irregularis* published to date, and made sure to specify where the data come from and how they were generated, to ensure that their usage in modelling is well-justified.

Reviewer #2

1.) Any thoughts as to why the fluxes from iAL1006 and the iRi1574 had a narrower range compared to other models? The result is mentioned around line 218 but never really explained.

To find an explanation for this observation, we investigate the coupling of the reactions in each of the models to the respective objective, i.e., the biomass reaction. We then classified the flux-carrying reactions into: (1) hard-coupled to the objective if the range of a reaction was equal to the range of the biomass reaction with a tolerance of 10^{-3} , (2) soft-coupled to the objective if the range was between zero and the range of the biomass reaction, (3) partially-coupled to the objective if the range was between the range of the biomass reaction and the maximum (i.e. 2000 mmol/gDW/h), and (4) uncoupled if the reaction range is equal to the maximum.

As a result, we found that both the iAL1006 and the iRi1574 model have the highest number of reactions that are hard- and soft-coupled to the biomass reaction, which provides an explanation for the narrow flux ranges at optimal biomass (Fig S3).

2.) Can the authors speak to the energy metabolism of their model (a critical detail in all models)? What ATP yield is produced from the input sugars?

We determined the ATP yield from all carbon sources (as single carbon sources) shown in Figure S3 by maximizing the flux through a sink reaction for cytosolic ATP while guaranteeing a growth rate of 50% or 0% of the predicted optimum (Fig S5). As a result, we observed that none of the carbon sources allowed for unlimited ATP production, which can happen when the models contain stoichiometrically-balanced cycles. The highest ATP production can be achieved from utilizing trehalose, while the highest ATP yield can be achieved from myristate. The high ATP production from trehalose can be explained by the fact that trehalose is a substrate of the biomass reaction. With trehalose not being growth-limiting, the excess trehalose that is not used for optimal growth can therefore be used for ATP production. The high yield predicted for myristate confirms experimental observation from Sugiura et al. (2020) ([10.1073/pnas.2006948117](https://doi.org/10.1073/pnas.2006948117)) that the ATP content increased 2.4-fold in the presence of myristate.

3.) The myristate results described around line 265 are somewhat confusing. The implication seems to be that myristate can replace a portion of the palmitate for fatty acid production? Is this correct?

Myristate cannot be elongated to produce palmitate as the required enzyme machinery is missing. However, it can restore biomass production via energy generation through beta-oxidation. We included the analysis to demonstrate the use of the model in replicating an observation from Sugiura et al. (2020) ([10.1073/pnas.2006948117](https://doi.org/10.1073/pnas.2006948117)), who found that the addition of myristate to the medium increases the biomass of the fungus. They also observed that the presence of myristate activates the expression of genes related to beta-oxidation, glyoxylate cycle, gluconeogenesis, and TCA cycle.

4.) Can the authors briefly explain in the manuscript why eMOMENT products strong correlations while FBA produces negative correlations with experimental data?

The addition to enzyme costs has been shown to increase the performance of constraint-based metabolic models. This is why we see a strong correlation with the eMOMENT approach while the results from FBA are rather unrealistic. Please note that the points for FBA and eMOMENT shown in Figure 2 are not completely mirrored, i.e., the relative increase or decrease in growth in comparison with the calculated growth rate are not exactly the opposite. The negative correlation may arise from the re-scaling of the biomass reaction with respect to the total protein content as well as from the usage of the enzyme constraints.

5.) In the study described around line 369, was transcriptomic data used instead of protein abundance data in the model, and can the authors justify that in these conditions the two datatypes are interchangeable?

We are not aware of a study that investigated absolute protein abundances of the different fungal structures. Therefore, we used transcript abundances instead as a proxy for protein abundance. We would like to point out that transcript abundances have been successfully applied to add constraints on reaction fluxes in constraint-based modelling in the past ([10.1371/journal.pcbi.1000489](https://doi.org/10.1371/journal.pcbi.1000489), [10.1111/tpj.12763](https://doi.org/10.1111/tpj.12763)), despite evident low correlations between transcript and protein abundances. We did not employ these approaches because we wanted to use a similar method as we did for prediction of growth with different carbon sources (i.e., eMOMENT) for comparability.

January 3, 2022

Prof. Zoran Nikoloski
University of Potsdam
Bioinformatics, Institute of Biochemistry and Biology
Karl-Liebknecht-Str. 24-25
Potsdam 14476
Germany

Re: mSystems01216-21R1 (Genome-scale modeling specifies the metabolic capabilities of *Rhizophagus irregularis*)

Dear Prof. Zoran Nikoloski:

Thank you for submitting your revised article and for carefully addressing the reviewers' comments. Your manuscript has been accepted, and I am forwarding it to the ASM Journals Department for publication. For your reference, ASM Journals' address is given below. Before it can be scheduled for publication, your manuscript will be checked by the mSystems senior production editor, Ellie Ghatineh, to make sure that all elements meet the technical requirements for publication. She will contact you if anything needs to be revised before copyediting and production can begin. Otherwise, you will be notified when your proofs are ready to be viewed.

Publication Fees:

We recognize that the video files can become quite large, and so to avoid quality loss ASM suggests sending the video file via <https://www.wetransfer.com/>. When you have a final version of the video and the still ready to share, please send it to mssystemsjournal@msubmit.net.

Sincerely,

Laura Hug
Editor, mSystems

Journals Department
Fig S5: Accept

Fig S3: Accept

Table S1: Accept

Supplementary Note: Accept

Fig S1: Accept

Supplemental Methods: Accept

Fig S4: Accept

Fig S6: Accept

Fig S2: Accept